# LANGUAGE-INDEPENDENT CROSS-LINGUAL CONTEXTUAL REPRESENTATIONS

## ABSTRACT

Contextual representation models like BERT have achieved state-of-the-art performance on a diverse range of NLP tasks. We propose a cross-lingual contextual representation model that generates language-independent contextual representations. This helps to enable zero-shot cross-lingual transfer of a wide range of NLP models, on top of contextual representation models like BERT. We provide a formulation of language-independent cross-lingual contextual representation based on mono-lingual representations. Our formulation takes three steps to align sequences of vectors: transform, extract, and reorder. We present a detailed discussion about the process of learning cross-lingual contextual representations, also about the performance in cross-lingual transfer learning and its implications.

## 1 INTRODUCTION

A cross-lingual text representation can generate language-independent representations: for pieces of text that are semantically similar, the produced representations are also similar, regardless of the language of the text. This provides a basis for dealing with multiple languages in NLP. Training a model on cross-lingual representations is a common approach for cross-lingual transfer learning (Huang et al., 2013; Artetxe & Schwenk, 2019).

To achieve desirable transfer learning performance, the cross-lingual representation model should be effective in two aspects: 1) the quality of text representation; and 2) the quality of cross-lingual mapping.

The current state-of-the-art text representation models are a family of pre-trained language model representations such as BERT (Devlin et al., 2019) and XLNet (Yang et al., 2019). These models produce contextual representations of text, i.e., a sequence of vectors corresponding to the sequence of text tokens. Each vector represents "a token in context," e.g., the semantics of a word in its local context and in the global context of the sentence.

Research on cross-lingual representation has so far been largely concentrated on word embeddings (Mikolov et al., 2013; Gouws et al., 2015; Ruder et al., 2017) and sentence embeddings (Schwenk & Douze, 2017; Artetxe & Schwenk, 2019). Word embeddings are static representations and are less expressive than contextual representations. Sentence embeddings perform respectably in some tasks but inferior in others, because there is information loss when too much information is crammed into a single vector (Conneau et al., 2018a). However, word embeddings and sentence embeddings are relatively easy to generalize to cross-lingual representations because they do not have dynamic inter-dependency among vectors.

We regard it as an interesting topic to investigate cross-lingual contextual representations. First, it is an interesting problem in its own to match two (or more) sequences of vectors, where each vector sequence has its order and locality relationships. Second, we hope to further improve cross-lingual transfer learning by leveraging the state-of-the-art text representation models.

In this study, we try to clarify the definition and characteristics of cross-lingual contextual representations, and propose a formulation to learn a cross-lingual contextual representations model based on existing mono-lingual pre-trained language models. We perform experiments to analyze the effect of the ingredients in our formulation and the final performance in transfer learning.

We also hope our investigation help to understand the nature of text representations in different languages: what are the commonalities between representations, how to cover the gaps, and ultimately what is transferable across languages with the help of state-of-the-art representation models.

**Related work**   Multilingual BERT learns a multi-lingual representation by training on a combined corpus of multiple languages. For dissimilar languages, the transfer performance is not satisfactory (Pires et al., 2019). XLM (Lample & Conneau, 2019) uses parallel corpus and a translational objective, and has achieved good transfer performance for many languages. However, XLM does not produce the same representation for different languages, so there is no guarantee of the performance in transfer learning. In multi-way neural machine translation (Firat et al., 2016) the situation is similar: multiple encoders produce the representation for each language, but because the representations are not language-independent, the decoder still needs to be trained on multiple language inputs.

A recent work on aligning contextual representations is (Schuster et al., 2019), which treats vectors in contextual representations as individual embeddings and learns a matrix to transform the embeddings space of different languages. The idea resembles the "transform" step in our formulation.

## 2   CROSS-LINGUAL CONTEXTUAL REPRESENTATIONS

First we define "language-independent cross-lingual contextual representations (CLCR)" and establish mathematical notations that we use in the following discussion.

### 2.1   DEFINITION

A cross-lingual contextual representation (CLCR) of text is a parameterized representation model $f(\theta) : (a_0, ..., a_L) \to (\boldsymbol{r}_0, ..., \boldsymbol{r}_N)$, where $a_0, ..., a_L$ is the sequence of text tokens in a sentence $s$, and $\boldsymbol{r}_0, ..., \boldsymbol{r}_N$ is the sequence of vectors produced by $f$ as the representation of $s$. A language-independent $f$ satisfies the following conditions:

1. (cross-lingual correspondence) if sentence $s^i$ in $i$-th language and sentence $s^j$ in $j$-th language are translations of each other, then by some distance metric $\mathcal{D}$ we have $\mathcal{D}(f(s^i; \theta), f(s^j; \theta)) < \delta$, where $\delta$ is a threshold.

2. (isometric property) if for some semantic metric $\mathcal{S}$, sentence $s^i$ in $i$-th language and sentence $s^j$ in $j$-th language satisfy $\mathcal{S}(s^i, s^j) < \gamma$, i.e., $s^i$ and $s^j$ are close in meaning, then by some distance metric $\mathcal{D}$ we have $\mathcal{D}(f(s^i; \theta), f(s^j; \theta)) < \delta(\gamma)$, where $\delta$ is a monotonic increasing function of $\gamma$.

3. (contextual representation) $\exists f_1, f_2, f_3 \ s.t. \ \boldsymbol{r}_i = f_3(f_2(a_i), f_1(a_{0...L\setminus i}))$. The vector $\boldsymbol{r}_i$ can roughly be interpreted as a representation of token $a_i$ in the context of the whole sentence.

Condition 1 and 2 ensure that CLCR produces similar representations for sentences with similar meaning, independent of language. Condition 3 is a characteristic of contextual representation models such as pre-trained language models like BERT.

### 2.2   EXISTENCE OF CLCR

To show that such cross-lingual contextual representation exists, we provide a formulation of CLCR based on mono-lingual contextual representations. We propose that one can generate CLCR with a three-step process. Starting from mono-lingual representations, each of the three steps corresponds to one problem that needs to be solved, in order to arrive at a cross-lingual contextual representation.

1. Alignment: the vectors of mono-lingual representation of different languages reside in different spaces. First we align the vector spaces $R^i$ with transformations $T_i : \boldsymbol{r}^i \to \boldsymbol{r}^C$, to transform into a common space $R^C$.

2. Granularity: different languages have different token granularities. Some language use several tokens to represent a word. Token alignment between translations $s_i$ and $s_j$ can be represented by a set $A = \{((a_m^i, ..., a_n^i), (a_p^j, ..., a_q^j))\}$, where tokens $(a_m^i, ..., a_n^i)$ and $(a_p^j, ..., a_q^j)$ are aligned "phrases."

Definition of **Finest Common Granularity**: find the token alignment $A^*$ that cannot be broken into a finer alignment. The finest common granularity for $s^i$ w.r.t. language $i$ and $j$ is defined as a grouping of the token sequence of $s^i$ in $A^*$: $FCG(s^i) = \{(a^i_0, ..., a^i_{n_0}), (a^i_{n_0+1}, ..., a^i_{n_1}), ..., (a^i_{n_m+1}, ..., a^i_L)\}$, where each element $(a^i_{n_i+1}, ..., a^i_{n_{i+1}}) \in A^*$.

In a nutshell, we want to group a series of tokens $(a^i_{n_0+1}, ..., a^i_{n_1})$ into a "phrase token" (key-point token), so that the key-point tokens have one-to-one correspondence between two languages. With key-point tokens derived from FCG, translations $s_i$ and $s_j$ have the same number of key-point tokens, and can then be represented by the same number of vectors.

3. Order: many different word orders exist in different languages, for example the SVO (subject-verb-object) and the SOV (subject-object-verb) order. We define an order mapping to map the orders of token from language $i$ to language $j$: $R_{i \to j} : m|s^i \to w$, where $((a^i_{n_m+1}, ..., a^i_{n_{m+1}}), (a^j_{n_w+1}, ..., a^j_{n_{w+1}})) \in A$. The order mapping $R_{i \to j}$ maps the position of a key-point token in the source sentence to the position of the corresponding key-point token in the target language.

To generate cross-lingual representations, we perform alignment transformation on the vectors of two mono-lingual representations, find their common granularity, and then reorder the vectors, to make the two representations similar.

## 2.3 Transfer learning with CLCR

Language-independent cross-lingual representations enables zero-shot transfer learning across languages. CLCR allows the most versatile models to be trained on cross-lingual representations. For a model $M$ taking contextual representations $(\boldsymbol{r}_0, ..., \boldsymbol{r}_N)$ as input, $M$ can be trained on dataset $\{(s^i, l)\}$ in language $i$, by first using CLCR $f$ to encode inputs: $M(f(s^i)) \to l$. In testing, model can perform inference on input in any language within $f$'s input languages.

## 3 Learning cross-lingual contextual representations

In this section, we detail our approach to learn CLCR based on the formulation in Section 2.2. We use two mono-lingual contextual representation models on language 1 and 2 to learn a cross-lingual contextual representation model.

## 3.1 Transform

In the first step, we learn a transformation $T$ parameterized by a multi-layer network:

$$T(\boldsymbol{r}) = Linear(\boldsymbol{r}) + Linear(Relu(Linear(\boldsymbol{r}))) \tag{1}$$

The network is trained to minimize the discrepancy between the transformed representation in language 1 and the representation in language 2: $D(T(\boldsymbol{r}^1_0, ..., \boldsymbol{r}^1_M)||(\boldsymbol{r}^2_0, ..., \boldsymbol{r}^2_N))$. Because we do not have alignment between the two sequences of vectors yet, we define $D$ with attention matching between the two sequences:

$$D(T(\boldsymbol{r}^1_0, ..., \boldsymbol{r}^1_M)||(\boldsymbol{r}^2_0, ..., \boldsymbol{r}^2_N)) = \sum_i \frac{1}{N} ||T(Attention(\boldsymbol{r}^2_i, (\boldsymbol{r}^1_0, ..., \boldsymbol{r}^1_M)) - \boldsymbol{r}^2_i|| \tag{2}$$

where $Attention$ is defined as using $\boldsymbol{r}^2_i$ as query to perform a weighted-sum of $(\boldsymbol{r}^1_0, ..., \boldsymbol{r}^1_N)$.

To train transformation $T$, we minimize the above loss $D$ on a parallel corpus $\{(s^1, s^2)\}$. We observed that straightforward gradient descent does not converge. We therefore used attention annealing during training to help our model converge to an optimal matching:

$$softmax(a_0, ..., a_N) = \frac{e^{a_0/T}}{\sum_i e^{a_i/T}} \tag{3}$$

where $T = 1/(1 + \beta \cdot step)$. As training progresses $softmax$ will gradually be replaced by $max$, which let the model find a one-to-one matching between the tokens. This is exactly required by the next step.

## 3.2 EXTRACT

In this step, we want to extract "key-points" from a sentence. These key-points are summarizations of local information (which may consist of multiple tokens), and the granularity of "key-points" is ideally consistent across different languages.

We extract key-points based on finest common granularity (FCG) between two languages: for a group of tokens $(a^i_{n_0+1}, ..., a^i_{n_1}) \in FCG(s^i)$, we use a single vector $\boldsymbol{r}_{n_0}$ to represent them. The number of vectors in the CLCR of $s^i$ is thus equal to $|FCG(s^i)|$.

Next, we train a network $P$ to predict key-points in $FCG(s^i)$. Although we can generate $FCG(s^i)$ by aligning parallel documents, in general CLCR should not rely on parallel documents and should predict key-points in a single sentence independently:

$$\boldsymbol{r}^i_k \text{ is the representation for key-point token } (a^i_{n_0+1}, ..., a^i_{n_1}) \tag{4}$$

$$\iff \exists! \, a^i_k \in \{a^i_{n_0+1}, ..., a^i_{n_1}\}, \, P(a^i_k) = 1 \tag{5}$$

The prediction network predicts one of the tokens in a key-point token to be the "pivot token." And the vector for that pivot token is regarded as the representation for the whole key-point token.

Finally, we need to generate ground-truth labels for training key-point prediction. In the definition above, the question "which word is the pivot token in a phrase?" do not have definite answers. Therefore, we train our representation model and let the model determine the pivot token in a key-point token. This is achieved by forced alignment via annealing in Section 3.1. After training the transformations $T$, we calculate the pairwise distance between $\boldsymbol{r}^1_i$ and $\boldsymbol{r}^2_j$: $D(T(\boldsymbol{r}^1_i)||\boldsymbol{r}^2_j) + D(T(\boldsymbol{r}^2_j)||\boldsymbol{r}^1_i)$, and use a threshold to find aligned pivot tokens.

Aligned pivot tokens (which is at the same time aligned key-point tokens) generated in this fashion closely resembles key-point tokens derived from FCG. As we do not have ground-truth word alignment of parallel corpus to generate real FCGs, we use our aligned key-points as pseudo-groundtruth alignments and use them as labels to train predictors in this and the next section.

One caveat of using parallel corpus is that translations are not unique. There can be multiple valid translations of a sentence. Thus labels derived from parallel corpus are intrinsically noisy and non-unique. We generate more reliable labels to train our prediction network, with a technique similar to (Liu & Tao, 2015).

## 3.3 REORDER

The aim of reordering is to make the order of contextual vectors $(\boldsymbol{r}^1_0, ..., \boldsymbol{r}^1_N)$ in language 1 compatible with the order in language 2. This can be achieved by learning an order prediction model, to predict the order mapping $R_{1\to2}$. We propose two methods for order prediction:

- Absolute order prediction: learn a prediction model to directly predict the position of token $a^1_i$ in language 2:

$$P(a^1_i) = R_{1\to2}(i) \tag{6}$$

Inference:

$$\underset{m_i \in Z^*}{\text{minimize}} \sum_i cost(P(a_i) = m_i) \; s.t. \; m_i \neq m_j \; \forall i, j \tag{7}$$

- Relative order prediction: learn a model to predict the relative order of two tokens $a^1_i$ and $a^1_j$ in language 2:

$$P(a^1_i, a^1_j) = sgn(R_{1\to2}(i) - R_{1\to2}(j)) \tag{8}$$

Inference:

$$\underset{m_i \in Z^*}{\text{minimize}} \sum_i \sum_{j>i} cost(P(a_i, a_j) = sgn(m_i - m_j)) \; s.t. \; m_i \neq m_j \; \forall i, j \tag{9}$$

The learned order prediction is $R_{1\to2} : i|s^1 \to m_i$. The vectors in language 1 can be reordered according to $R_{1\to2}$, so that the reordered vectors have the order of language 2. The reordered vector sequence is $(\boldsymbol{r}^1_{R_{2\to1}(0)}, ..., \boldsymbol{r}^1_{R_{2\to1}(N)})$, where $R_{2\to1}$ is the reverse mapping of $R_{1\to2}$.

## 4 EXPERIMENTS

In this section, we perform experiments to learn language-independent cross-lingual contextual representation based on the approach in Section 3. We also perform transfer learning and analysis experiments to examine the characteristics of CLCR.

### 4.1 LEARNING

We sequentially examine the details and the effectiveness of each of the three steps in our formulation to generate CLCR: transform, extract, and reorder.

The basic experiment setup is: we use mono-lingual pre-trained language models from Google BERT (Devlin et al., 2019), for English and Chinese[1]. The 12-layer BERT-base model is used. We use parallel corpus from The United Nations Parallel Corpus v1.0 (Ziemski et al., 2016). Experiments are performed with Pytorch (Paszke et al., 2017).

**Transform**    As shown in Table 1, training transformation $T$ to minimize distance $D$ with attention annealing managed to converge to a low distance between the two representations. Lower distance indicates that a better alignment is found (because unaligned and misaligned tokens always contribute to high distance). And a good alignment is also conducive to learning a better transformation.

Table 1: Distance $D$ achieved by transformation $T$

| Model | $D(T(s^1)\|\|s^2)$ | $D(T(s^2)\|\|s^1)$ |
|---|---|---|
| Linear Regression | 0.055 | 0.078 |
| Attention (w/o annealing $T = 1$) | 0.138 | 0.207 |
| Attention (with annealing $\beta = 0.001$) | 0.028 | 0.048 |
| Attention (with annealing $\beta = 0.01$) | **0.025** | **0.037** |

During training, we alternately optimize the representation model for language 1 and 2, so that they do not learn a trivial solution of producing the same constant representation. The learning curve is shown in Figure 1.

The produced alignment is used as labels in the next two steps.

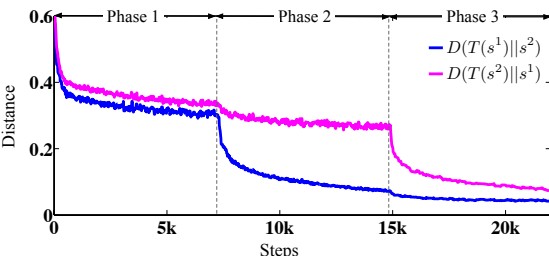

Figure 1: Learning curve of distance $D$. Training phase 1 (left): fix both model, learn $T$ only. Training phase 2 (middle): finetune language 1 representation model with annealing. Training phase 3 (right): finetune language 2 representation model with annealing.

**Extract**    A binary classifier is trained to predict whether a token is a pivot token of a key-point. An example prediction is illustrated in Figure 2. It can be observed that only tokens possess significant meaning is predicted as pivot tokens. For languages (e.g., Chinese) that use multiple tokens to represent a word, only one token per word is predicted as pivot tokens. Considering the labels are

---

[1]https://github.com/google-research/bert

pseudo-groundtruth produced from only one translation per sentence, the overall prediction accuracy is respectable (see Table 2).

Using confident labels slightly improves prediction accuracy, and makes the prediction significantly less ambiguous. This is very important for producing consistent predictions of key-points, such that transfer learning models on top of CLCR can be trained on a consistent representation.

Table 2: Key-point prediction accuracy

| Model | Accuracy | | Precision | | Recall | |
|---|---|---|---|---|---|---|
| | en | zh | en | zh | en | zh |
| Finetune | 0.859 | 0.823 | 0.886 | 0.854 | 0.936 | 0.847 |
| Confident labels | **0.864** | **0.826** | 0.875 | 0.838 | 0.959 | 0.872 |

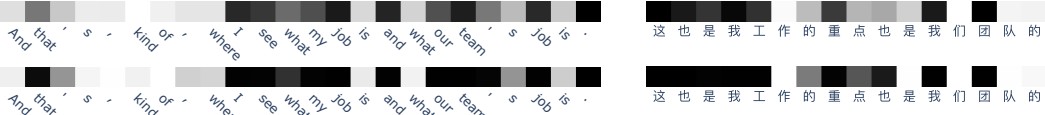

Figure 2: An example of predicted key-points, with (below) and without (above) confident labels

**Reorder** In absolute order prediction, we use an N-way classifier to predict the absolute position of a token. In relative order prediction, a binary classifier is used to predict the relative order of a pair of tokens. During inference, beam search is used to find the best order assignment of tokens, so that the total cost is minimized. We simply use the probability $p$ of the classifier output as the cost of a particular assignment.

Table 3 reports the reordering performance of two methods with different beam size $n$. We calculate 2-gram to 4-gram BLEU with respect to the groundtruth order. Relative order prediction performs much better than absolute order prediction, getting 2-gram predictions correct over half of the time and 4-gram predictions correct nearly one-third of the time.

Table 3: Reorder performance

| Model | BLEU (2-gram) | | BLEU (3-gram) | | BLEU (4-gram) | |
|---|---|---|---|---|---|---|
| | en | zh | en | zh | en | zh |
| Absolute order prediction ($n = 1$) | 0.309 | 0.386 | 0.166 | 0.227 | 0.101 | 0.152 |
| Absolute order prediction ($n = 10$) | 0.331 | 0.393 | 0.189 | 0.239 | 0.121 | 0.165 |
| Relative order prediction ($n = 1$) | 0.535 | 0.566 | 0.384 | 0.410 | 0.290 | 0.308 |
| Relative order prediction ($n = 10$) | **0.582** | **0.624** | **0.419** | **0.468** | **0.316** | **0.368** |

The example in Figure 3 illustrates that absolute order prediction suffers from ambiguity except near the beginning of the sentence, because the exact position depends on the specific translation and therefore does not have a unique correct answer. Relative order prediction is much more robust because it is invariant to the location of the pair of tokens in the sentence, the classifier only makes predictions based on the relative order relation between them.

## 4.2 CROSS-LINGUAL TRANSFER

**Natural Language inference** XNLI (Conneau et al., 2018b) is a cross-lingual natural language inference corpus consisting of test examples in 15 languages. We use the English and Chinese test set in our evaluation. We use MultiNLI (Williams et al., 2018), an English NLI dataset as the training set to train the task model.

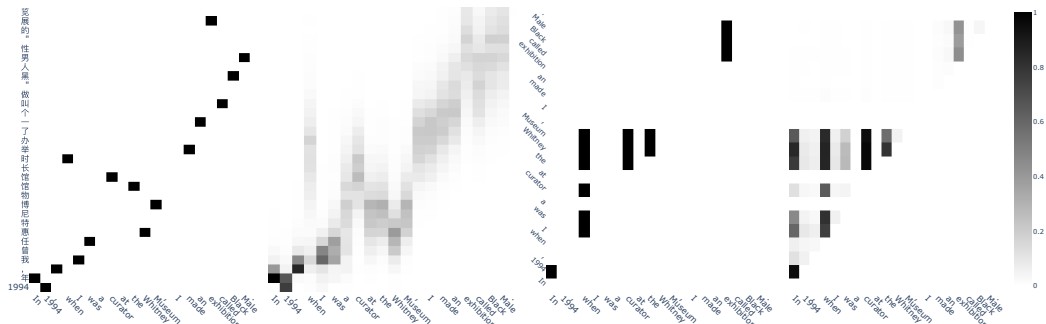

Figure 3: A example of order prediction. From left to right: (1) ground-truth absolute order. (2) predicted absolute order. (3) ground-truth relative order. (4) predicted relative order.

To evaluate the performance of our cross-lingual contextual representation model, we train the ESIM model (Chen et al., 2017) with representation produced by CLCR as input. An advantage of CLCR over multi-lingual pre-trained language models is that complex task models can be trained on top of the representation. To examine the effect of each of the three steps in our formulation of CLCR, we also experimented with different combinations of steps.

Our full CLCR model (T+E+R) achieved a transfer accuracy of 71.7%, much better than multilingual BERT and even better than translating test set to English. The transfer gap is on par with state-of-the-art models (Table 4).

It is clear that CLCR works best when all three steps are present. Without extraction and reordering, model on top of CLCR does not perform as well. This is because the ESIM model makes use of order information of the input sequence, the number and the order of vectors should match for successful zero-shot transfer to another language.

It is also observed that extraction and reordering lower performance on English by roughly 1%. The performance loss is expected, and not beyond acceptable range. During the training of CLCR, the English representation model is finetuned to accommodate another language, and during imperfect extraction and reordering some information is inevitably lost.

We also set an upper bound for the transfer performance of CLCR by using pseudo-groundtruth alignments (T+pGA). Pseudo-groundtruth alignments give the "correct" extraction and reordering, and indicates the performance of the best possible T+E+R model. The remaining transfer gap (6.4%) is thus intrinsic to the task (variation of test set difficulty) and the mono-lingual representation models (different performance of BERT-base-en and BERT-base-zh).

Table 4: Performance of zero-shot cross-lingual transfer. T, E, R stands for Transform, Extract, and Reorder, respectively. pGA stands for pseudo-groundtruth alignment. Gap stands for the transfer gap from English to Chinese (the drop of accuracy in percentage)

| Model | XNLI test set | | |
| --- | --- | --- | --- |
| | en Acc. | zh Acc. | Gap |
| *CLCR* | | | |
| T+pGA | 81.2 | **74.8** | **6.4** |
| T | 81.2 | 68.0 | 13.2 |
| T+E | 80.6 | 69.3 | 11.3 |
| T+E+R | 80.2 | **71.7** | **8.5** |
| *Non-CLCR models* | | | |
| Multilingual BERT | 81.4 | 63.8 | 17.6 |
| Translate train | 81.4 | 74.2 | 7.2 |
| Translate test | 81.4 | 70.1 | 11.3 |
| XLM | 85.0 | **76.5** | **8.5** |

Although the performance of our CLCR on Chinese XNLI is inferior to XLM, the transfer gap is roughly the same. XLM uses much more data and training time than our approach, and thus starts with better performance than BERT-base on English. Also XLM is not language-independent, which means it is not feasible to use XLM as representations to learn models for cross-lingual transfer. One can only fine-tune XLM itself on a language and hope that other languages also benefit from fine-tuning.

### 4.3 ANALYSIS

**Edge probing** To investigate the effect of transformation, extraction, and reordering, we measure what kind of information is retained in CLCR by comparing the representation before and after training. We use a technique called Edge Probing (Tenney et al., 2019b). In edge probing, a series of tasks are used as "probes" to measure the syntactic and semantic information contained in representations. The jiant toolkit (Wang et al., 2019) is used to perform the experiments.

Comparing CLCR with the original BERT-base English model in Table 5, the transformation step (T) generally lowers scores of most probes by a small fraction. This indicates while most information remains in the representation after training, there is a certain portion of in-

Table 5: Edge probing statistics of CLCR

| | F1 score | | |
|---|---|---|---|
| | BERT | T | T+E |
| POS | 97.6 | 92.4 | 82.2 |
| Consts. | 86.8 | 80.8 | 75.4 |
| Deps. | 95.4 | 92.8 | 84.5 |
| Entities | 96.1 | 94.8 | 90.9 |
| SRL | 89.3 | 86.0 | 83.1 |
| Coref. | 95.8 | 92.0 | 91.5 |
| SPR | 83.4 | 83.1 | 82.6 |
| Relations | 77.7 | 75.9 | 75.9 |

compatible syntactic and semantic representations across languages. That information is lost and explains the performance loss on XLI on the source language (English). Extraction (E) further lowers scores, but mainly due to failing to predict labels on non-pivot tokens. Because the probing model used in (Tenney et al., 2019b) is an attention pooling-based order-less model, reordering (R) does not alter the performance on these probing tasks.

**Training cost and beyond** A total of 250,000 pairs of parallel sentences are used in this study. This is less than 2% of the data used to train XLM. Increasing the size of parallel corpus does not lead to further improvement. Also the total training time of CLCR, including the three steps is less than 5 hours, on a single GPU. This indicates that with good mono-lingual contextual representations like BERT, learning cross-lingual alignment of representations could be a relatively easy task, without the need for extensive supervision and training. Under our formulation, the performance of CLCR is built upon and also limited by the quality of mono-lingual representations. But this more of an opportunity than a problem because parallel corpus is always the more limited resource.

## 5 CONCLUSION

In this paper we investigated language-independent cross-lingual contextual representations. We proposed a three-step process to learn cross-lingual contextual representations from mono-lingual representations. In each of the three steps, an intuitive method is employed with the aim of letting the model itself learn to ultimately align two representations. Experiments confirmed that state-of-the-art pre-trained language models are powerful enough to learn to align two languages with moderate supervision. This could indicate that with models like BERT now it is easier than ever to cover the gap in cross-lingual transfer learning.

Our formulation provides a new approach to combine contextual representations with arbitrary task models for zero-shot cross-lingual transfer learning. We experimented with the NLI task and the ESIM model, but the method could in principle be applied to a wide range of tasks and models.

With the advancement of pre-trained language models and their availability, we hope to extend the approach to more languages and more powerful models. We also hope our investigation could inspire further insight into the nature of language-independent representations and cross-lingual transfer learning.

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

# A  METHOD OVERVIEW

We provide an overview diagram to show the whole process of generating language-independent cross-lingual contextual representations, using an example in Figure 4.

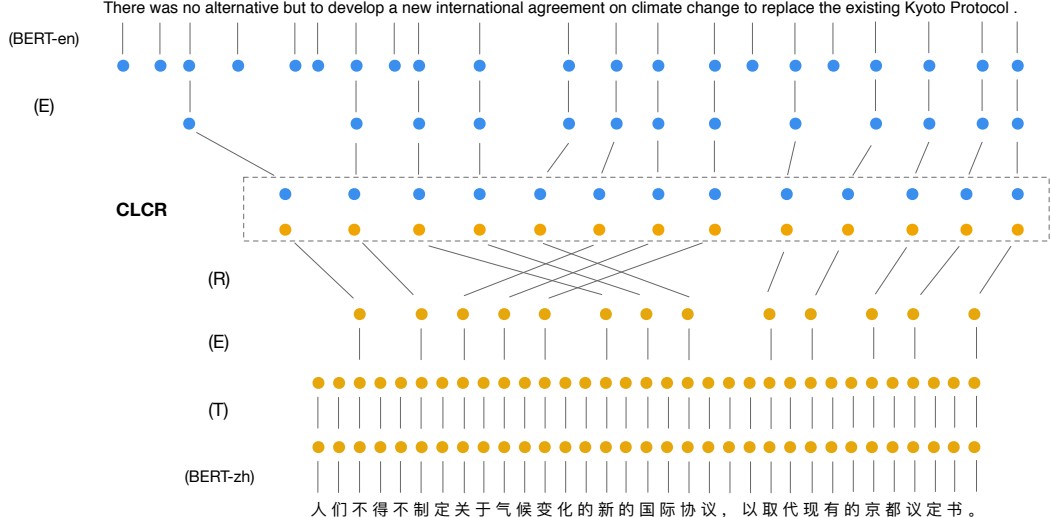

Figure 4: An example illustrating CLCR in inference. For a pair of language (in this example, English and Chinese), one can decide to perform transformation and reordering on one side, and perform extraction on both sides. After training of the transformation (T), extraction (E) and reordering (R) model, for input text in one of the two languages, a sequence of operation is performed: Embed-T-E-R or Embed-E depending on the language. The generated sequence-of-vector representation (in the box with dashed line) is the same for text in both languages, assuming perfect T, E, and R models.

# B  ZERO-SHOT CROSS-LINGUAL TRANSFER

Language-independent cross-lingual contextual representations enable zero-shot cross-lingual transfer for a variety of NLP models. For example, with learned CLCR between English and Chinese, one can train a task model with Embed-E on English, then perform inference with Embed-T-E-R on Chinese (Figure 5).

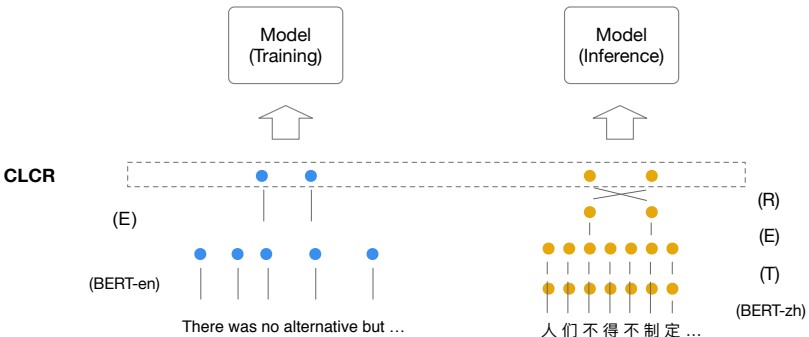

Figure 5: An example of zero-shot cross-lingual transfer with CLCR

## C  MODEL DETAILS AND MORE ANALYSIS

In our experiments, we use the ninth layer output of the 12-layer BERT-base model as mono-lingual contextual representations. Based on the observation in (Tenney et al., 2019a), middle-high layers of transformer language models typically exhibit good semantic properties. The English and Chinese BERT-base models we used are identical except the language of the corpus used to trained the model. We also experimented with using different pre-trained language models, for example ERNIE (Sun et al., 2019) for Chinese (BERT-base is still used for English), but the cross-lingual transfer performance is inferior. This shows that for best aligning of representations in CLCR, the two mono-lingual representation models should share the same structure and objective in pre-training.

Table 6: Edge probing statistics

|  | BERT-base | | | T | | | T+E ($\delta = 0.2$) | | | T+E ($\delta = 0.5$) | | |
|---|---|---|---|---|---|---|---|---|---|---|---|---|
|  | F1 | Pre. | Rec. | F1 | Pre. | Rec. | F1 | Pre. | Rec. | F1 | Pre. | Rec. |
| POS | 97.6 | 97.3 | 97.8 | 92.4 | 91.7 | 93.2 | 82.2 | 91.5 | 74.6 | 77.6 | 92.8 | 66.7 |
| Consts. | 86.8 | 86.8 | 86.8 | 80.8 | 82.4 | 79.2 | 75.4 | 80.2 | 71.1 | 71.5 | 80.1 | 64.6 |
| Deps. | 95.4 | 96.2 | 94.7 | 92.8 | 94.1 | 91.5 | 84.5 | 90.4 | 79.2 | 78.8 | 89.4 | 70.5 |
| Entities | 96.1 | 96.6 | 95.6 | 94.8 | 95.5 | 94.2 | 90.9 | 95.3 | 86.8 | 87.8 | 95.3 | 81.4 |
| SRL | 89.3 | 91.5 | 87.1 | 86.0 | 89.0 | 83.1 | 83.1 | 88.8 | 78.0 | 80.8 | 88.6 | 74.2 |
| Coref. | 95.8 | 95.8 | 95.8 | 92.0 | 90.8 | 93.2 | 91.5 | 91.5 | 91.5 | 89.4 | 89.4 | 89.4 |
| SPR | 83.4 | 84.2 | 82.6 | 83.1 | 83.3 | 82.9 | 82.6 | 84.5 | 80.8 | 82.5 | 82.3 | 82.6 |
| Relations | 77.7 | 85.5 | 71.2 | 75.9 | 85.4 | 68.2 | 75.9 | 84.1 | 69.2 | 74.8 | 84.3 | 67.3 |

In Table 6, we list more detailed statistics in probing the cross-lingual representations. The drop of score in the extraction (E) step is closely linked to the threshold $\delta$ used in key-point prediction. Threshold $\delta$ controls the number of key-points predicted in a sentence. A higher $\delta$ means a smaller number of tokens are predicted as key-points. This results in better accuracy in matching the granularity of two languages by only predicting more confident key-points, but fewer key-points means more tokens are discarded and thus more information is potentially lost. This becomes obvious by comparing the recall of T and T+E models in probing tasks. We empirically chose a trade-off value of $\delta = 0.2$ in all of our experiments.

We need to note that in E step the reduction of recall in probing tasks does not necessarily mean reduced performance of NLP models on top of CLCR. Significant decrease of recall is mainly observed on part-of-speech, constituent and dependency labeling tasks. These tasks have dense labels, and labels on non-key-point tokens are ignored by CLCR after step E. However, in training to align sentences the representation model is forced to summarize local information in the representation of key-points, so that information from non-key-point tokens could still be available for task model on top of CLCR.

