# OpenReview forum: "Language-independent Cross-lingual Contextual Representations"
_ICLR.cc/2020/Conference — Reject_

### Official Review · AnonReviewer3 · 2019-10-18
**Official Blind Review #3**

**Rating:** 3

**Review:**

This paper proposes a new approach to learning contextualised cross-lingual representations based on a three-step procedure called: transform-extract-reorder. The proposed model is tested on zero-shot cross-lingual transfer for one language pair: English-Chinese, where the gains are reported over the initial multilingual BERT model which does not rely on any bilingual signal. I like the main modeling idea, and the actual model is described quite nicely and in an intuitive way. However, I believe that much more work is needed in terms of proper experiments including: a) additional strong and insightful baselines; b) experiments with more language pairs; c) experiments across other transfer tasks. The paper also does not include nor compares to all relevant previous work. My main comments and remarks are as follows:

*Related work and Novelty* The topic of learning contextualised cross-lingual representations has become quite popular since the rise of ELMo and BERT. However, the authors cite only one paper that proposed such contextualised xling representations (the work of Schuster of al.). However, they never compare to that model, which is quite weird given the fact that the main goal of all xling representation models is to enable cross-lingual transfer in the first place. A comparison to the cited work of Schuster et al., plus additional comparisons to other (non-cited) models are required to clearly understand the main empirical benefits of the proposed three-step framework. Other relevant papers which are not cited nor compared against: Mulcaire et al. (NAACL 2019); Aldarmaki and Diab (NAACL 2019).

The authors should really stress the key novelty of their approach in comparison to the existing literature (which is mostly based on simpler projection-based methods).

- Note that there are more papers on the subject getting published soon such as Mulcaire et al. (CoNLL 2019) and Liu et al. (CoNLL 2019); while making comparisons to models introduced in those papers is not required, it would be nice to also briefly mention that latest work in the related work section once it gets published.

*Comparisons to XLM* Based on the limited set of results reported in the paper, it seems that the proposed model cannot match the performance of the XLM model (Lample and Conneau, NeurIPS 2019). The authors try to justify the usage of their method over XLM by emphasising the fact that their method produces language-independent xling representations, which XLM cannot do. However, there are multiple issues here: 1) it is not clear why exactly this requirement of language-independence is needed for a task such as zero-shot XNLI; 2) the paper does not demonstrate the usefulness of that language-independence property empirically at all. To me, it just seems like an unsuccessful attempt to make a conceptual distinction between the proposed model and XLM given the fact that XLM seems to significantly outperform the proposed model.

Page 8 ("Training cost and beyond"). The authors claim that 250k pairs of parallel sentences is enough to learn a near-optimal cross-lingual model and that the model saturates with more parallel sentences. I don't see it as a positive aspect of the model as claimed by the authors. This means that the model cannot optimally encode knowledge about variety of cross-lingual contexts during its training, i.e., it stops learning earlier than expected. Why does that happen? A plot showing transfer learning results of XLM versus the proposed model versus some static word embedding model that also exploits parallel data would be very useful. The authors also do not even speculate why their model saturates already with 250k sentence pairs. Given the limited set of results (only one task and only language pair) it also remains unknown how general this phenomenon is and whether the same point is hit with some other language pairs and with other tasks. This definitely requires further and more thorough investigation.

*Other Comparisons and Experiments*
As mentioned, the paper is very limited when it comes to proper thorough evaluation. Most experiments focus on intrinsic/internal evaluations of different steps of the proposed framework, or some probing tests, which can also be seen simply as diagnostic experiments. The actual downstream experiments are conducted on only one language pair and for only one (zero-shot) transfer task. This is definitely not sufficient to draw any generalisable claims, and it prevents us to dig deeper into other interesting aspects of the model. I would suggest the authors to maybe run the model across the same range of tasks as done in the XLM work of Lample and Conneau, and definitely for more languages (ideally diverse target languages).

Also, besides XLM and translation-based baselines reported by multilingual BERT's github page, there is actually no other baselines, which really makes it hard to put this work in context, and understand its usefulness. For instance, static cross-lingual word embeddings could be added to ESIM and used to evaluate on XNLI as well (see, e.g., the work of Glavas et al. (ACL 2019)). It would be interesting to report the benefits of replacing such static vectors with truly contextualised representations. Also, as mentioned before, comparisons to other models that learn contextualised cross-lingual representations are definitely something that should be included in the paper.

*Other Comments*
- Based on the result from Table 4, it seems that, in order to enable fully language-independent representations, one must sacrifice some of the monolingual performance, as the numbers drop from T over T+E to T+E+R variant. Is the same pattern visible for other languages monolingually? Why does this happen?
- As mentioned before, the whole emphasis is of language-independence is somewhat oversold in the whole paper without providing sufficient empirical evidence that this is crucial for transfer performance.

**Experience Assessment:**

I have published in this field for several years.

**Review Assessment: Checking Correctness Of Derivations And Theory:**

I assessed the sensibility of the derivations and theory.

**Review Assessment: Checking Correctness Of Experiments:**

I carefully checked the experiments.

**Review Assessment: Thoroughness In Paper Reading:**

I read the paper thoroughly.

---

### Official Review · AnonReviewer2 · 2019-10-22
**Official Blind Review #2**

**Rating:** 3

**Review:**

This paper presents a new method to obtain cross-lingual contextual embeddings by aligning monolingual ones through 3 steps: transform each token individually, merge them as needed to obtain a uniform granularity across languages, and reorder them.

While I think that the proposed method has some interest, and the extensive ablation experiments are useful to better understand its behavior, I do not think that the it makes enough merits to be accepted in the conference. I feel that the paper tends to overly complicate things, and it is often difficult to extract any clear idea from it. The proposed method is also much more complicated than previous approaches, yet it does not perform better than them (XLM has better absolute results and exactly the same cross-lingual transfer gap, while being substantially simpler). More concretely:

- The paper tends to overly complicate things. For instance, Section 2.1 and 2.2 try to mathematically formalize very basic intuitions. Unless the formalization is important for clarity (which is not, as these are obvious ideas) or necessary later in the paper (which is not either), there is no point in doing that. This only makes the paper more difficult to follow than what it should.

- The only extrinsic evaluation is in XNLI, where the authors evaluate the zero-shot cross-lingual transfer performance from English into Chinese. However, the proposed method does not bring any improvement over the current state-of-the-art in this setup. The proposed method gets 80.2% and 71.7% accuracy in English and Chinese, respectively, leaving an absolute transfer gap of 8.5%, while the XLM model from Lample and Conneau (2019) obtains substantially better results (85.0% and 76.5%) with the exact same transfer gap of 8.5%. This could still be good enough if the proposed method had some other advantage over the previous SOTA, but I cannot find any and, in contrast, I do find some disadvantages (see below).

- In addition to being more complicated than previous approaches, the proposed method also introduces new hyperparameters and seems more difficult to train. For instance, the authors need to incorporate annealing to train the transformation module, and the model seems quite sensitive to the corresponding hyperparameter (Table 1).

- While both multilingual BERT and XLM simply fine-tune a pre-trained BERT model to perform some downstream task, the proposed method is used as a feature extractor, and the authors train an ESIM (LSTM) model on top. The reported experiments do not control this factor (i.e. what would happen if one learns an ESIM model on top of XLM)?

- The authors highlight that their system can be trained in "less than 5 hours, on a single GPU", while "XLM uses much more data and training time", but this is quite deceptive. Your approach also requires training a monolingual BERT model for each language, which is even more expensive than training a single joint model as XLM does. It is true that one could potentially use publicly available monolingual models, but most pre-trained models in languages other than English are already multilingual, anyway, so I do not see a strong practical justification for this.

- The proposed model and the ones it is compared to do not use the same training data. This can have some justification (it might be computationally prohibitive for the authors to pre-train their own models, which might be the reason why they use public models trained on different data) but they should be more upfront about this. In relation to this, it is unfair to remark that the proposed method uses less parallel data than XLM, while not even mentioning that it uses more monolingual data (if I am not wrong, XLM was only trained in Wikipedia, while BERT also used a book corpus at least for English). To make things worse, the authors claim that "XLM uses much more data and training time than our approach", which seems wrong.

- The authors criticize XLM because it "does not produce the same representation for different languages, so there is no guarantee of the performance in transfer learning". This might be true, but is it anyhow different for your proposed method? Your method is not better empirically, and it does not have any theoretical guarantee either.

- As the authors themselves acknowledge, the proposed method is similar in spirit to Schuster et al. (NAACL'2019) -which is also much simpler- but they do not compare to it in their experiments.

All in all, I think that the paper tends to overly complicate things, and ultimately fails to answer a simple central question: why should one prefer your approach over previous methods like XLM, or what is it that it makes it otherwise interesting or relevant?

**Experience Assessment:**

I have published in this field for several years.

**Review Assessment: Checking Correctness Of Derivations And Theory:**

I assessed the sensibility of the derivations and theory.

**Review Assessment: Checking Correctness Of Experiments:**

I assessed the sensibility of the experiments.

**Review Assessment: Thoroughness In Paper Reading:**

I read the paper at least twice and used my best judgement in assessing the paper.

---

### Official Review · AnonReviewer4 · 2019-10-22
**Official Blind Review #4**

**Rating:** 3

**Review:**

This paper proposes a method to learn language-independent cross-lingual contextual representations by mapping the representations of a monolingual model in one language to the representations of a monolingual model in another language. The proposed approach consists of three steps: 1. A transformation is learned that minimizes the distance between the contextual word representations of the two models of a sentence and its translation in a parallel corpus. 2. Each sentence is summarized as a sequence of key-point tokens based on phrase alignment. 3. The contextual word vectors are reordered based on an order prediction model.
The authors perform experiments on intrinsic tasks and on XNLI, mapping between an English and a Chinese BERT model. They outperform multilingual BERT (mBERT) on the latter.

Overall, the approach is novel, but the steps seem overly complicated. The extrinsic evaluation is the weakest point of the paper. Because of this, I tend to a Weak Reject. I would be willing to increase my score if additional languages are added to the evaluation and if the steps are better motivated and compared to simpler alternatives.

The high-level steps of the approach (transform, align, reorder) make sense. They seem to be inspired by classical phrase-based MT pipelines, but this connection is not made clear. In particular, some of the steps seem unnecessarily complicated and I am wondering whether the authors tried or compared to simpler alternatives. As a parallel corpus is used in the first step, word alignment could be automatically obtained by FastAlign without the use of attention matching or using a phrase table that is learned in an unsupervised way as in recent work in NMT (https://arxiv.org/pdf/1804.07755.pdf). For the second step, embeddings of aligned phrases could be averaged or the head of a phrase could be used instead of predicting key-points with a separate network.

The data requirements between the steps seem somewhat inconsistent. A parallel corpus is used in the first step, but explicitly not used in the second step. While I agree that language-independent cross-lingual representations should not use a parallel corpus, it would have been good if the first step could have also been performed without one (or with a smaller size).

The extrinsic evaluation of the paper could be improved. The accuracy on each intrinsic step is evaluated. Without any reference or comparison, I found it hard to tell how good these numbers are so this evaluation did not add much for me. The main extrinsic evaluation of the paper is on XNLI but only employs an English and a Chinese model. When training cross-lingual models, the aim is to train approaches that not only work for one or two languages but for many. In light of this, I find it hard to tell from the results on a single language pair how well the approach will generalize to other languages, particularly as mBERT's performance on zh XNLI is comparatively weak (https://arxiv.org/pdf/1901.07291.pdf). There are other publicly available BERT models such as for German (https://deepset.ai/german-bert) on which the approach could be tried. If compute is a concern, then the approach could be applied to ELMo representations. This would also enable a comparison to Schuster et al. (2019; https://arxiv.org/abs/1902.09492).

Finally, the data requirement of 250k parallel sentences is prohibitive for many language pairs where this approach would be valuable. It would be good to see a chart on how model performance develops with the number of parallel sentences to see if a smaller number of parallel sentences would be viable in practice.

**Experience Assessment:**

I have published one or two papers in this area.

**Review Assessment: Checking Correctness Of Derivations And Theory:**

N/A

**Review Assessment: Checking Correctness Of Experiments:**

I carefully checked the experiments.

**Review Assessment: Thoroughness In Paper Reading:**

I read the paper thoroughly.

---

### Author Response · Authors · 2019-11-15
**Author response to reviews**

Thanks for the comments from all reviewers!

We acknowledge that one of the main weaknesses of this paper is in its evaluation, which is only performed on a single pair of language and on a single task. We are working on more experiments to strengthen the evaluation:

- Evaluation on more languages, such as German

- Evaluation on more tasks, especially more complex tasks such as reading comprehension. Although NLI is the most commonly used task (recently) to evaluate cross-lingual representations, it does not fully take advantage of the language-independent aspect of our proposed method. While the successful zero-shot transfer of ESIM model partially demonstrated the use of language-independent cross-lingual representations, evaluation on more complex NLP tasks could reveal situations where the language-independency of representation is more critical to zero-shot transfer performance.

Unfortunately, we are not able to update on new results here yet. We really appreciate the suggestions by the reviewers which we would take to further improve our work.

---

### Decision · Program_Chairs · 2019-12-19

**Decision:**

Reject

**Comment:**

The paper proposes a method to learn cross-lingual representations by aligning monolingual models with the help of a parallel corpus using a three-step process: transform, extract, and reorder. Experiments on XNLI show that the proposed method is able to perform zero-shot cross-lingual transfer, although its overall performance is still below state-of-the-art jointly trained method XLM.

All three reviewers suggested that the proposed method needs to be evaluated more thoroughly (more datasets and languages). R2 and R4 raise some concerns around the complexity of the proposed method (possibly could be simplified further). R3 suggests a more thorough investigation on why the model saturates at 250,000 parallel sentences, among others.

The authors acknowledged reviewers' concerns in their response and will incorporate them in future work.

I recommend rejecting this paper for ICLR.